# Down-Syndrome-Related Maternal Dysbiosis Might Be Triggered by Certain Classes of Antibiotics: A New Insight into the Possible Pathomechanisms

**DOI:** 10.3390/antibiotics12061029

**Published:** 2023-06-08

**Authors:** Gábor Ternák, Gergely Márovics, Katalin Sümegi, Zsolt Bánfai, Gergely Büki, Lili Magyari, András Szabó, Béla Melegh

**Affiliations:** 1Institute of Migration Health, Medical School, University of Pécs, Szigeti út 12., H-7624 Pécs, Hungary; 2Department of Public Health Medicine, Medical School, University of Pécs, Szigeti út 12., H-7624 Pécs, Hungary; 3Department of Medical Genetics, Medical School, University of Pécs, Szigeti út 12., H-7624 Pécs, Hungary; 4Department of Biochemistry and Chemistry, Medical School, University of Pécs, Szigeti út 12., H-7624 Pécs, Hungary

**Keywords:** Down syndrome (DS), microbiome, dysbiosis, antibiotics, tetracycline, penicillin

## Abstract

Down syndrome (DS) is a leading human genomic abnormality resulting from the trisomy of chromosome 21. The genomic base of the aneuploidy behind this disease is complex, and this complexity poses formidable challenges to understanding the underlying molecular basis. In the spectrum of the classic DS risk factor associations, the role of nutrients, vitamins, and, in general, the foodborne-associated background, as part of the events ultimately leading to chromosome nondisjunction, has long been recognized as a well-established clinical association. The integrity of the microbiome is a basic condition in these events, and the dysbiosis may be associated with secondary health outcomes. The possible association of DS development with maternal gut microbiota should therefore require more attention. We have hypothesized that different classes of antibiotics might promote or inhibit the proliferation of different microbial taxa; and hence, we might find associations between the use of the different classes of antibiotics and the prevalence of DS through the modification of the microbiome. As antibiotics are considered major disruptors of the microbiome, it could be hypothesized that the consumption/exposure of certain classes of antibiotics might be associated with the prevalence of DS in European countries (N = 30). By utilizing three different statistical methods, comparisons have been made between the average yearly antibiotic consumption (1997–2020) and the estimated prevalence of people living with DS for the year 2019 as a percentage of the population in European countries. We have found strong statistical correlations between the consumption of tetracycline (J01A) and the narrow-spectrum, beta-lactamase-resistant penicillin (J01CF) and the prevalence of DS.

## 1. Introduction

Down syndrome (DS) is the most common genomic disorder of intellectual disability and is caused by a trisomy of the human chromosome 21 (HSA21). The presence of a supernumerary chromosome 21 results in a collection of clinical features commonly known as DS. Individuals suffering from DS are also more likely to develop different health conditions, including hypothyroidism, autoimmune diseases, obstructive sleep apnea, epilepsy, hearing and vision problems, hematological disorders, including leukemia, recurrent infections, anxiety disorders, early onset Alzheimer’s disease (AD), and congenital heart diseases, such as atrioventricular septum defect, patent ductus arteriosus, and tetralogy of Fallot [1]. DS occurs in all populations and countries, but differences in maternal age at conception between regions and ethnicities influence the number of live births. In Europe, DS prevalence in live births has been decreasing slightly since the 1990s, although there are substantial regional differences [2]. The signs of DS usually occur during prenatal development [3]. The average live birth with Down syndrome for 100,000 delivery, estimated by the WHO [4] and calculated for 1997–2020 in 24 European countries is 99.872, but considerable differences could be observed (the lowest is Austria: 15.735, the highest is Latvia: 215.433). There are well-documented, classic events in the pathogenesis of the DS, such as maternal age, chromosome nondisjunction, and the influence of paternal factors, all of which are results of early DS research. Unanswered questions have also been addressed by methods in the recent genomic medicine era, such as robust genome-wide association studies, which have been performed to identify the genomic susceptibility factors behind the mother’s predisposition. However, it is still clear that there are numerous factors outside of the human genome influencing DS susceptibility in pregnancy.

The role of the human microbiome in health and diseases had been widely documented in the literature. Several factors are capable of modifying the composition of the human microbiome, resulting in dysbiosis, and as a consequence, the development of different diseases is reported in publications [5,6].

The most effective microbiome modifiers are antibiotics, which can enter humans (animals) as therapeutic agents, or from the environment as the result of antibiotic pollution.

The “growth-promoting” effect of the antibiotics was first observed in the late 1940s, when chicks were accidentally fed with culture media used for cultivating antibiotic-producing organisms and they started growing faster than others. Experiments later confirmed this observation [7,8]. The growth-promoting effect of antibiotics could not have been reproduced in germ-free animals [9]. It was concluded that the “collateral” effect of antibiotics might appear through the presence of bacterial flora. This observation led to the industrial utilization of antibiotics, mixed with animal fodder to enhance the growth of food animals, and later, it also facilitated the emergence of polyresistant microbes [10].

In normal circumstances, intestinal flora starts developing quickly after birth. Humans are colonized by highly adapted microbiota with coevolved functions that promote human health, development, and disease resistance. Several studies have identified *Bifidobacterium* and *Bacteroides* species maternally transmitted to infants, but the disruption of maternal transmission by cesarean section and antibiotic exposure around birth is associated with a higher incidence of pathogen colonization and immune-related disorders in children, such as allergies, asthma, etc. [11].

Microbes from the placenta, amniotic fluid, and umbilical cord blood provide a diverse spectrum of bacterial exposures to the developing fetus [12]. Shortly after birth, or even before, the neonatal gut is rapidly colonized by facultatively anaerobic bacteria, typically including strains of Enterobacter, *Enterococcus*, Staphylococcus, and Streptococcus genera [13,14]. Breastfeeding appears to have long-term effects on the microbiome and its effects on the immune system and the gastrointestinal (GI) tract [15].

The human GI tract contains relatively rich and complex microbial communities in healthy individuals. Intestinal microbes harbor genes that encode thousands of microbial enzymes and metabolites. Microbes colonize the human body and shift in composition as humans age, with a gradual increase in microbial diversity during childhood and relative stabilization during adolescence and adulthood. The amount of the microbiome might be calculated as 2–3% of the body weight. The recently estimated number of bacterial cells in the microbiome in the 70 kg “reference man” is 3.8 × 10^13^ [16].

New technologies have permitted scientists to phylogenetically determine and/or quantify the components of the gut microbiota by analyzing nucleic acids (DNA and RNA) directly extracted from stools. Most of these techniques are based on the extraction of DNA and the amplification of the 16S ribosomal RNA gene (rRNA) [17,18]. 16S rRNA sequencing has become the most useful technique to highlight the diversity and abundance of the microbiome. The 16S rRNA gene sequences can be exploited with polymerase chain reaction (PCR) and metagenomics sequencing to characterize the microbial strains [19]. The gut microbiome encodes over three million genes, producing thousands of metabolites, whereas the human genome consists of approximately 23,000 genes [20]. Gut microbiota is composed of several species of microorganisms, including bacteria, yeast, and viruses. Taxonomically, bacteria are classified according to phyla, classes, orders, families, genera, and species. Only a few phyla are represented, accounting for more than 160 species. The dominant gut microbial phyla are *Firmicutes*, *Bacteroidetes*, *Actinobacteria*, *Proteobacteria*, *Fusobacteria*, and *Verrucomicrobia*, with the two phyla *Firmicutes* and *Bacteroidetes* representing 90% of gut microbiota. The *Firmicutes* phylum is composed of more than 200 different genera such as *LactoBacillus*, *Bacillus*, *Clostridium*, *Enterococcus*, and *Ruminicoccus*. *Clostridium* genera represent 95% of the *Firmicutes* phyla. *Bacteroidetes* consist of predominant genera such as *Bacteroides* and *Prevotella*. The *Actinobacteria* phylum is proportionally less abundant and mainly represented by the *Bifidobacterium* genus [21]. Detailed analyses indicated that certain bacterial taxa, called core microbiomes, are always present in adults. The core microbiome is composed of *Bacteroides*, *Eubacterium*, *Faecalibacterium*, *Alistipes*, *Ruminococcus*, *Clostridium*, *Roseburia*, and *Blautia*; with *Faecalibacterium prausnitzii*, *Oscillospira guillermondii*, and *Ruminococcus obeum* as the top three taxa shared by all adults [22].

The gut microbiota influences the health of the host. It provides sufficient benefits in the form of immune system development, the prevention of infections, nutrient acquisition, and perhaps even brain and nervous system functionality [23].

Even though microbiomes might vary among healthy individuals, microbial functions are well-conserved and associated with the generation of microbial metabolites, which influence both microbes and hosts. It might be assumed that the modification of the microbiome results in the production of different molecules, which might induce the development of different diseases through the gut–brain axis (GBA), or other mechanisms. The imbalance of the human microbiome is called dysbiosis, which is the change in abundance and the diversity of the microbial taxa of the intestine. This association makes dysbiosis a central concept for understanding how the human microbiota contributes to health and disease [24,25]. The gut–brain axis involves bidirectional interplay between the brain and the gut, including the gut-associated immune system, the enteric neuroendocrine system (ENS), and the gut microbiome. This new view of gut–brain interactions has explained the pathophysiology of several brain disorders that had previously been attributed exclusively to pathophysiological processes limited to the brain. Research in the past two decades indicated disorders of altered GB interactions as well as psychiatric and neurological disorders such as depression, anxiety, AD, Parkinson’s disease (PD), and autism spectrum disorder (ASD) [26], even if the molecular background is not appropriately identified. In an average European population, even within a country, disease-related dysbiosis might occur among the people eating a similar diet. In this case, the microbiome alteration develops as a result of other external factors, such as antibiotic consumption/pollution (or other). The pathogenic role of dysbiosis is well-documented in publications when fecal microbial transplantation (FMT) was applied to ameliorate the symptoms of different diseases (Pakinson’s disease, autism, ulcerative colitis, etc.) [27].

Bacterial metabolites produced by the microbiome can also mediate host processes and functionally complement host metabolic capabilities. A recent review article by Ruan et al. summarized the bacterial molecular products, which are known as quorum sensors, influencing bacterial homeostasis, growth, spore formation, programmed cell death, virulence, and biofilm formation. Microbes can produce biologically active compounds including, but not limited to, gamma-aminobutyric acid (GABA), tryptophan metabolites, polyamines, and histamine. Microbial neuromodulators such as GABA can communicate with the enteric and central nervous systems, and microbial-derived immunomodulators such as histamine can interact with intestinal immune cells. Serpins, which are microbial-derived immunomodulators, are similar to eukaryotic serine protease inhibitors and they suppress inflammatory responses by inhibiting elastase activity. *LactoBacillus*-secreted lactocepins are bacterial enzymes that can degrade pro-inflammatory signals such as lymphocyte-recruiting chemokine and can suppress pro-inflammatory signaling cascades. Short-chain fatty acids (SCFAs) are implicated in immune regulation, pH regulation, sodium and water absorption, etc. The most abundant and well-studied SCFAs are acetate, propionate, and butyrate; however, the intestinal composition of SCFAs is contingent on microbial composition, diet, and intestinal pH. Outer-membrane vesicles (OMVs) are another key immunomodulatory factor which the gut microbiota produces [28].

Dysbiosis has been observed in several diseases, and the permanent question is whether dysbiosis is a consequence or an association of the development of the disease. Clinical observations and experiments indicated that dysbiosis could be observed long before the clinical manifestation of the disease. Autism spectrum disorder (ASD) seems to be related to an abnormal composition of the gut microbiome, indicating that *Clostridia*, *Desulfovibrio*, *Sutterella*, and *Bacteroidetes* are elevated in the stool of autistic children. In contrast, *Firmicutes*, *Prevotella*, and *Bifidobacteria* have been noted to be decreased [29,30,31]. The gut microbiota seems to also play a role in the development and progression of obesity. Germ-free mice that receive fecal microbes from obese humans gain more weight than mice that received microbes from healthy-weight humans. A large study of UK twins found that the genus *Christensenella* was rare in overweight people, and when given to germ-free mice, prevented weight gain [32]. A higher ratio of *Firmicutes* to *Bacteroides*/*Prevotella* in obese people enhances the microbial genes involved with the degradation of polysaccharides and increases the level of SCFAs [33]. Appropriate evidence suggests that a shift in the gut microbiome (GM) composition may play an important role in the pathogenesis of PD by facilitating the characteristic ascending neurodegenerative spread of α-synuclein aggregates from the enteric nervous system to the brain [34].

The knowledge of the role of the gut microbiome in modulating brain function has rapidly increased over the past decade, but mainly in animal models. Increasing clinical and preclinical studies implicate the microbiome as a possible key susceptibility factor for neurological disorders, including Alzheimer’s disease, autism spectrum disorder, multiple sclerosis, PD, and stroke [35].

Antibiotics are considered major disruptors of the microbiome. The gut flora of newborn babies and young children is particularly sensitive to antibiotic exposure. Studies have reported that preterm infants who received prolonged antibiotic treatment had less diverse bacterial populations and reduced species richness in their gut and more antibiotic-resistance genes [36]. A detailed review of the literature on the effects of antibiotics influencing the gut flora can be found in reference [37,38,39].

It is important to notice that, when we discuss the association between the use of antibiotics and the development of certain diseases, that different classes of antibiotics might promote or inhibit the proliferation of different microbial taxa and hence, support or inhibit the development of different diseases through the modification of the gut flora via the gut–brain axis. A comparative analysis of the consumption of different classes of antibiotics and the prevalence/incidence of certain metabolic, neurodegenerative diseases, and malignancies in European countries showed a significant association between antibiotic consumption patterns and the prevalence/incidence of different diseases. It has been estimated that in countries with an outstanding consumption of narrow-spectrum penicillin and tetracycline, the incidence/prevalence of type I diabetes (T1DM), PD, multiple sclerosis (MS), and ASD, along with certain malignancies (melanoma, breast cancer, colorectal cancer) are higher than in other countries. Countries with a high consumption of broad-spectrum antibiotics recorded a high prevalence of childhood obesity, AD, and type II diabetes (T2DM) [40].

The alteration of the gut microbiome in DS patients has also been reported [41,42,43]. Recent papers have indicated that pregnant mothers delivering babies with DS showed an altered microbiome compared to the control group. *Clostridiaceae*, *Parteurellacae*, and *Pastereullaes* were more abundant in DS pregnancies, compared to the controls in this pilot study [43], and this alteration might occur in the newborn babies as well, even if this was not investigated in these studies.

## 2. Objectives/Hypothesis

Based on the above-detailed background, we have hypothesized that the alteration of the maternal microbiome might be a factor inducing the development of DS. Antibiotics are considered a major disruptor of the human microbiome, entering humans as therapeutic agents or from the environment, as pollutants. The effect of antibiotics on the gut flora is well-known [37]. We have suspected that different classes of antibiotics might promote or inhibit the proliferation of different microbial taxa and hence, we might find associations between the use of the different classes of antibiotics and the prevalence of DS through the modification of the microbiome.

## 3. Methods

We calculated the average yearly antibiotic consumption in the Defined Daily Dose/1000 inhabitants/day (DID) between 1997 and 2020 of different classes of antibiotics, covering 94% of the total systemic antibiotic consumption in the community featured in the ECDC database [44], Comparison was performed with the estimated prevalence of people living with DS for the year 2019 in 30 European countries [45].

## 4. Statistics

Statistical tests were performed using IBM SPSS 26 software package. Pearson correlation was applied to estimate the correlation between antibiotic consumption and the prevalence of DS. A significant correlation (positive/negative) was considered when *p* values were ≤0.05. A nonsignificant correlation was estimated when the *p* values fell between 0.051–0.09. Positive (supportive) and negative (inhibitor) significant correlations were considered and evaluated. Ordinal logistic regression analysis was performed to determine the odds ratio and confidence intervals (OR, CI 95%) for each antibiotic class. The Kruskal–Wallis test was performed to determine statistically significant differences among the groups formed by country ranking. A significant difference was considered when *p* values were ≤0.05, and a nonsignificant difference was considered when *p* values were between 0.051–0.09.

## 5. Results

Statistical results are included at the bottom line of Table 1 and the significant associations are marked with yellow color. Related diagrams are shown in Figure 1 and Figure 2. We have observed a significant positive association between the consumption of tetracycline (J01A, Pearson *p*: 0.007, OR *p*: 0.005, Kruskal–Wallis *p*: 0.034), narrow-spectrum, beta-lactamase-resistant penicillin (J01CF, Pearson *p*: 0.077, OR *p*: 0.021, Kruskal–Wallis: 0.152 /nonsignificant/), and the prevalence of DS in 30 European countries. A strong, significant correlation has been detected with all three statistical methods regarding the use of tetracycline and the prevalence of inhabitants living with DS, while the use of the narrow-spectrum penicillin showed a nonsignificant correlation with the prevalence of DS using Pearson correlation. The OR *p* indicated an elevated risk of DS with the use of narrow-spectrum penicillin, but the Kruskal–Wallis test did not find any association between the databases. This difference is well demonstrated in Figure 1 and Figure 2 as the diagram showing the association between the use of tetracycline and DS is steeper than the one indicating the association between the use of penicillin and DS.

## 6. Discussion

Almost 95% of DS is from a full trisomy of chromosome 21. The remaining cases are due to either mosaicism for chromosome 21 or the inheritance of a structural rearrangement leading to a partial trisomy of the majority of its content. Full trisomy 21 and mosaicism are not inherited but originate from errors in cell divisions during the development of the egg, sperm, or embryo. In addition, full trisomy for chromosome 21 should be further divided into cases of maternal origin, the majority, and cases of paternal origin, less than 10% [46]. Advanced maternal age at conception is considered a major risk factor for trisomy 21, as is true for all human autosomal trisomies [1]. Females in advanced age (above 35 years) are at higher risk of infertility, miscarriage, or having a pregnancy affected by congenital birth defects such as Down syndrome. However, the molecular background of nondisjunction is still being investigated, and the outcome of the genomic research shows that multiple exogenous and endogenous factors contribute to the age-related increase in oocyte aneuploidy. Some research associates specific gene variations with Down syndrome. It is suspected that the shortage of folate has been shown to disrupt the way DNA and chromosomes separate and recombine. In the case of such a gene variation in one of the parents that interfere with their body’s capability to metabolize folate may enhance their chances of having a child with Down syndrome [47]. Researchers found folate regulator gene mutations that are associated with maternal meiosis II error. Smoking is also considered a risk factor, and in an Asian population with the habit of using smokeless chewing tobacco, the risk of delivering babies with DS was higher [48]. Analysis of the mother’s reproductive tract flora and the intestinal flora of DS cases indicated significant differences regarding the intestinal microbiome of DS patients compared to a healthy control group, while the reproductive tract flora did not show any significant difference [49]. The composition of the microbiome in children with DS was 16.67% *Bacteroides*, 37.35% *Firmicutes*, 26.13% *Proteobacteria*, 50% *Verrucomicrobia*, and 15.64% *Actinobacteria*. The normal group was mainly composed of 45.30% *Bacteroides*, 22.85% *Firmicutes*, 15.64% *Proteobacteria*, 28% *Verrucomicrobia*, and 3.02% *Actinobacteria*. An analysis of the fecal samples of women who delivered babies suffering from DS indicated that the microorganisms of the families *Clostridiaceae* and *Pasteurellaceae* were more abundant in the group of women having delivered DS neonates than the group of women having delivered healthy newborns. The analysis of the collateral effect of antibiotics on the gut microbiome indicated that tetracycline/doxycycline reduced the abundance of several commercial components of the microbiome isolates, most effectively, *E. coli*, even at low concentrations [50]. Interestingly enough, the use of narrow-spectrum penicillin and tetracycline might also be associated with the prevalence/incidence of certain neurodegenerative diseases, such as PD and MS [51,52,53]. Maternal microbial dysbiosis has been implicated in adverse postnatal health conditions in offspring, such as obesity, cancer, and neurological disorders, but in the case of DS, most of the available literature is focused on maternal risk factors, and the only certain risk factors for the birth of a child with DS are advanced maternal age at conception and recombination errors, even though the molecular mechanisms leading to chromosome 21 nondisjunction are still a matter of debate. Trisomy of chromosome 21 (TS21) is the most common autosomal aneuploidy compatible with postnatal survival with a prevalence of 1 in 700 newborns. Its phenotype is highly complex with constant features, such as mental retardation, dysmorphic traits, and hypotonia, and variable features, including heart defects, susceptibility to Alzheimer’s disease (AD), type 2 diabetes, obesity, and immune disorders. Overexpression of genes on chromosome 21 (Hsa21) is responsible for the pathogenesis of Down syndrome (DS) phenotypic features, either directly or indirectly, since many Hsa21 genes can affect the expression of other genes mapping to different chromosomes. Many of these genes are involved in mitochondrial function and energy conversion and play a central role in mitochondrial dysfunction and chronic oxidative stress, consistently observed in DS subjects [54].

Tetracycline is a broad-spectrum polyketide antibiotic produced by the Streptomyces genus of *Actinobacteria* and it specifically inhibits the 30S ribosomal subunit, blocking the binding of the aminoacyl-tRNA to the acceptor site on the mRNA-ribosome complex on the bacterial cell. When this process stops, a cell can no longer maintain proper functioning and will be unable to grow or further replicate. The tetracyclines enter the bacterial cell wall in two ways: passive diffusion and an energy-dependent active transport system, which is probably mediated in a pH-dependent manner. Protein synthesis is ultimately inhibited, leading to a bacteriostatic effect. In light of recent evidence that tetracycline binds to various synthetic double-stranded RNAs (dsRNAs) of random base sequences, it is suggested that the double-stranded structures may play a more important role in the binding of tetracycline to RNA rather than specific base pairs, as was previously thought. This is imperative to consider possible alternative binding modes or sites that could help explain the mechanisms of action of the tetracycline against various pathogens and disease conditions [55].

The narrow-spectrum, beta-lactamase-resistant penicillin (J01CF) inhibits the proliferation of Gram-positive cocci, *Bifidobacteria*, *Lactobcilli*, *Eubacteria*, and *Lachnospiraceae* in the microbiome, and promotes the proliferation of *Enterobacteria* and *Bacteroidaceae* [37].

The association of the antibiotic-modified microbiome and the development of consecutive, noncontagious diseases, prompted researchers to suspect that the use of antibiotics might influence the microbiome to reverse the clinical manifestations of certain diseases through the possible elimination of intestinal pathogens triggering the disease [56,57,58].

Clinical solutions to modify gut microbiomes generally focus on depleting the overabundant microbial taxa or overall microbial load using antibiotics or antifungal agents. Modulation through diet or supplementation with live microbes (single or mixed species) might be a possibility, but the introduction of fecal microbial transplantation (FMT) has been found as a better solution. Antimicrobial drugs are not generally considered appropriate for the long-term management of chronic conditions, given the need for repeat dosing and concerns about the emergence of antimicrobial resistance. Early data indicate that the use of antimicrobial drugs, as a means to eliminate pathogenic microbiomes before FMT or microbial supplementation, improves the engraftment of beneficial species and enhances treatment efficacy [59].

The new antibiotic-treatment method for AD and PD has emerged as a possible novel approach to ameliorate the symptoms in those cases and other ailments. The research in animal experiments AD/PD models provided sufficient evidence of the anti-amyloidogenic, anti-inflammatory, antioxidant, and antiapoptotic activity of tetracyclines, associated with cognitive improvement. The discovery of the neuroprotective effects of minocycline and doxycycline in animals initiated an investigation of their clinical efficacy in AD and PD patients, leading to inconclusive results because long-stating therapy with tetracycline and minocycline might raise safety issues. The safety issues should be considered as the long-lasting use of antibiotics might result in the emergence of resistant pathogens and the modification of the microbiome with unforeseeable consequences. The sub-antimicrobial doxycycline doses should be carefully explored for their effectiveness and long-term safety, especially in AD/PD populations. Minocycline may have some neuroprotective activity in various experimental models such as cerebral ischemia, traumatic brain injury, amyotrophic lateral sclerosis, PD, Huntington’s disease, and MS [60,61].

Autism spectrum disorder (ASD) has a complex range of neurodevelopmental symptoms, including the impairment of social interactions and communication, together with restrictive and repetitive patterns of behaviors. According to the latest studies, autistic children have shown significant gut microbiota composition changes and the GI symptoms may represent the inflammatory processes. It has been reported that microbial interventions, such as probiotics and antibiotics (vancomycin) can contribute to the reduction of social behavioral symptoms and the level of inflammation in individuals with ASD [62,63].

Alteration of the gut–brain axis has been reported in people with MS, suggesting a possible role in disease pathogenesis and making it a potential therapeutic target. Antibiotic treatment ameliorated experimental autoimmune encephalitis (EAE) in mice through effects mediated via the gut microbiota. Disease amelioration was not observed when antibiotics were given intraperitoneally, bypassing the gut, suggesting that the modulation of the gut microbiota produces protective effects.

A recent report found that oral treatment with ampicillin decreased EAE severity, but in different EAE models, the results were controversial [64].

Aging is associated with dysbiosis, defined as a loss of number and diversity in gut microbiota, which has been linked with various aspects of cognitive functions. Therefore, the gut microbiome has the potential to be an important therapeutic target for symptoms of cognitive impairment. It was reported that 12 weeks of probiotic treatment with a supplement containing *LactoBacillus* acidophilus, *LactoBacillus casei*, *Bifidobacterium bifidum*, and *LactoBacillus bifidum* positively affected the cognitive function and metabolic homeostasis in AD patients [65].

Limitations of Our Study: Our results could not be applied at the individual level, and we could not provide data on the molecular mechanism of how antibiotics act in the development of trisomy.

Strength of the Study: Using international, large-scale data analysis and three different statistical methods, this study provides convincing evidence of the significant involvement of certain classes of antibiotics inducing dysbiosis, which might lead to the development of DS.

## 7. Conclusions

Our comparative analysis using three different statistical methods clearly indicated a strong statistical association between the average yearly consumption (1997–2020) of tetracycline (J01A) and narrow-spectrum, beta-lactamase-resistant penicillin (J01CF) expressed in DID and the prevalence of people living with DS in 30 European countries estimated for 2019. It is suspected that antibiotics (J01A, J01CF)-modified gut microbiome could play some role in the development of DS-related dysbiosis in the mothers delivering babies with DS, and this altered flora also affects the intestinal microbiome of the newborn babies with DS. Although this statistical association could not be applied to the individual level, a theory might be raised that certain molecules produced by the altered maternal microbiome could act as a factor for enhancing the development of chromosome 21 trisomy, which is a possible new insight in the pathomechanisms of DS, but further research is necessary to confirm this possible etiology. It could be theorized that probiotics/prebiotics, taken for an appropriately long period before the planned pregnancy, might stabilize the maternal microbiome, preventing DS-related dysbiosis.

## Figures and Tables

**Figure 1 antibiotics-12-01029-f001:**
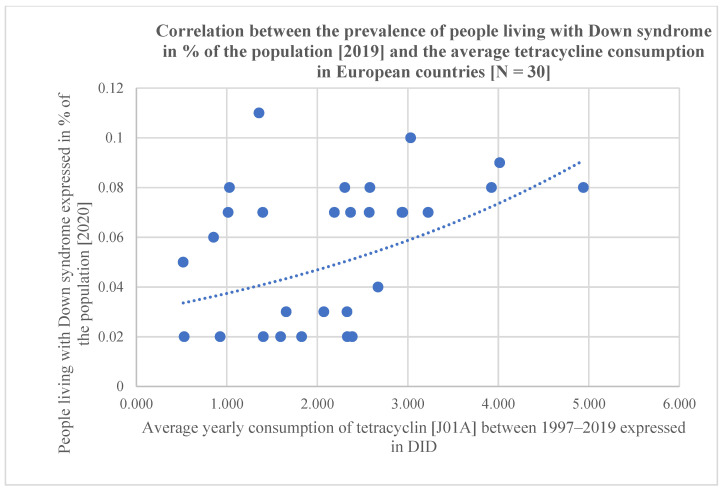
Scatter diagram showing the positive association between tetracycline consumption (J01A) and the prevalence of people living with DS, estimated for 2019 (in 30 European countries).

**Figure 2 antibiotics-12-01029-f002:**
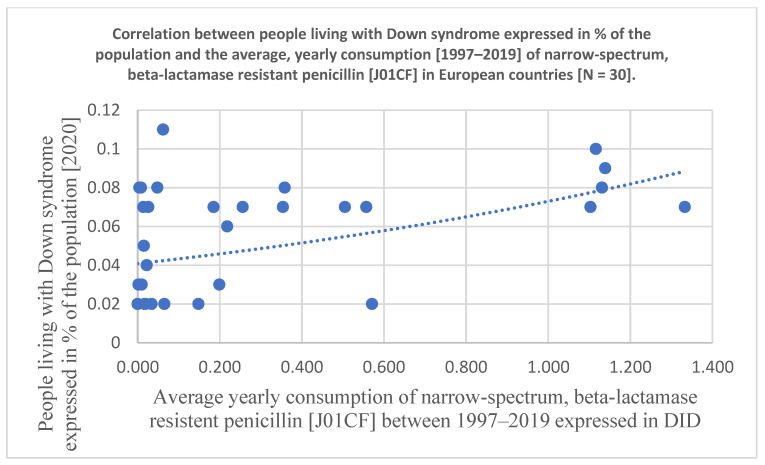
Scatter diagram showing the association between the consumption of narrow-spectrum, beta-lactamase-resistant penicillin (J01CF) and the prevalence of people living with DS estimated for 2019 (In 30 European countries).

**Table 1 antibiotics-12-01029-t001:** Average, yearly consumption of different classes of antibiotics between 1997–2020 expressed in DID compared to the prevalence of DS in the population expressed in %. Significant positive correlations are marked with yellow color, and a nonsignificant positive association is marked with blue. Pink color marks the DS data and the statistical results.

Antibiotic Consumption 1997–2020 DID	J01	J01A	J01C	J01CA	J01CE	J01CF	J01CR	J01D	J01F	J01M	People Living with DS in % of the Population, 2019	%
Austria	11.683	1.03	4.251	0.819	0.946	0.008	2.474	1.565	3.076	1.311	Austria	0.08
Belgium	21.427	2.368	8.725	3.796	0.091	0.256	4.578	2.387	3.205	2.206	Belgium	0.07
Bulgaria	17.828	2.386	6.55	4.039	0.953	0.019	1.56	2.672	2.589	1.991	Bulgaria	0.02
Croatia	17.911	1.404	7.581	2.467	1.007	0.034	4.074	3.172	2.641	1.496	Croatia	0.02
Cyprus	27.338	3.226	9.454	3.118	0.101	0.026	6.234	5.923	3.126	4.623	Cyprus	0.07
Czech Rep.	14.685	2.333	5.539	1.355	1.877	0.065	2.222	1.55	3.028	1.03	Czech Rep.	0.02
Denmark	14.068	1.398	8.782	2.732	4.575	1.103	0.378	0.03	2.031	0.455	Denmark	0.07
Estonia	10.947	2.072	3.701	2.422	0.264	0.01	1.007	0.892	1.947	0.83	Estonia	0.03
Finland	16.157	3.925	4.559	2.349	1.514	0.048	0.65	2.094	1.474	0.799	Finland	0.08
France	24.478	3.224	11.423	6.804	0.18	0.354	4.067	2.818	4.087	1.923	France	0.07
Germany	12.61	2.574	3.419	2.033	1.064	0.014	0.302	1.908	2.293	1.205	Germany	0.07
Greece	30.474	2.581	8.579	4.088	0.441	0.004	4.018	7.183	8.026	2.686	Greece	0.08
Hungary	14.711	1.595	5.269	1.467	0.634	0	3.164	2.162	3.054	1.843	Hungary	0.02
Iceland	19.148	4.941	9.219	3.392	2.446	1.131	2.232	0.567	1.56	0.797	Iceland	0.08
Ireland	18.256	3.032	8.219	2.816	0.93	1.116	3.374	1.57	3.413	0.858	Ireland	0.1
Italy	21.524	0.518	9.092	3.509	0.013	0.015	5.56	2.835	4.726	3.073	Italy	0.05
Latvia	10.653	2.329	4.04	2.717	0.096	0.002	1.226	0.544	1.435	0.982	Latvia	0.03
Lithuania	16.714	1.656	8.744	5.103	2.225	0.199	1.227	1.301	1.747	0.983	Lithuania	0.03
Luxembourg	22.41	2.189	7.974	3.09	0.088	0.185	4.618	4.087	3.998	2.413	Luxembourg	0.07
Malta	18.465	1.356	6.265	0.537	0.092	0.062	5.572	4.034	3.771	2.1	Malta	0.11
Netherlands	9.218	2.304	2.954	1.266	0.357	0.358	0.972	0.089	1.383	0.836	Netherlands	0.08
Norway	15.046	2.944	6.114	1.919	3.681	0.505	0.009	0.144	1.567	0.468	Norway	0.07
Poland	18.773	2.671	6.186	3.653	0.421	0.022	2.092	2.315	3.346	1.276	Poland	0.04
Portugal	18.094	1.015	7.836	2.006	0.028	0.557	5.248	2.308	3.179	2.496	Portugal	0.07
Romania	22.897	0.926	10.725	4.063	0.695	0.571	5.478	4.281	2.866	2.954	Romania	0.02
Slovakia	21.098	1.827	8.279	2.232	2.82	0.017	3.218	3.604	4.632	1.842	Slovakia	0.02
Slovenia	12.873	0.529	7.15	2.148	1.982	0.148	2.881	0.56	2.462	1.265	Slovenia	0.02
Spain	17.697	0.854	9.166	3.611	0.101	0.218	5.134	2.089	2.538	2.359	Spain	0.06
Sweden	13.37	2.936	6.387	1.047	3.821	1.333	0.182	0.28	0.73	0.825	Sweden	0.07
UK	15.294	4.016	5.851	3.257	0.746	1.139	0.734	0.546	2.627	0.534	UK	0.09
2019 R (Pearson)	0.174	0.479	−0.040	−0.139	−0.043	0.328	0.005	0.072	0.109	−0.009		
2019 *p* (Pearson)	0.358	0.007	0.835	0.464	0.821	0.077	0.981	0.706	0.567	0.963		
2019 OR	1.074	3.896	0.861	0.741	0.497	14.491	0.942	0.510	2.031	2.737		
2019 CI 95%	0.932–1.246	1.632–11.626	0.560–1.304	0.418–1.262	0.193–1.117	1.764–186.274	0.572–1.508	0.139–1.578	0.784–5.969	0.537–16.944		
2019 OR *p*	0.324	0.005	0.478	0.274	0.109	0.021	0.806	0.263	0.160	0.239		
2019 Kruskal–Wallis *p*	0.431	0.034	0.348	0.49	0.237	0.152	0.842	0.897	0.517	0.362		

## Data Availability

Not applicable.

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
