# Peer review of "Down-Syndrome-Related Maternal Dysbiosis Might Be Triggered by Certain Classes of Antibiotics: A New Insight into the Possible Pathomechanisms"

_antibiotics, 2023, doi:10.3390/antibiotics12061029_

Round 1

Reviewer 1 Report

From the scientific point of view, this work provides a statistical analysis where it is possible to establish a correlation between the consumption of antibiotics with a molecular mechanism associated with the disruption of the microbial cell wall synthesis process. However, there are several elements that must be considered

The differences found between the prevalence between countries is an element to consider, since it would make it possible to identify a sample with the genetic, epigenetic and phenotypic characteristics that could contribute to identifying the causes and molecular bases that allow to delve into the disease etiology. Although this question is alluded to in the work “There are well-documented, classic events in the pathogenesis of the DS like maternal age, chromosome nondisjunction, and influence of paternal factors, all of which are results of early DS research. Unanswered questions have also been addressed by methods in the recent genomic medicine era, like robust genome-wide association studies which have been performed to identify the genomic susceptibility factors behind the mother’s predisposition. However, it is still clear that there are numerous factors outside of the human genome influencing DS susceptibility in pregnancy”. I understand that it should be properly referenced

One of the strong criticisms that must be made of this type of work refers to the term dysbiosis, which alludes to an abnormal qualitative and quantitative configuration of the microbiome. In this sense, it is worth reminding the authors that this term It is clinical and from the point of view of Microbial Ecology it does not make sense since the variation of species and the quantitative fluctuations in the microbiome are constant, therefore, there is not a situation of absence of dysbiosis, but rather there is a gradient of variation that is constant and more or less intense. This variation is supposed to have as a final result the production of certain secondary metabolites that elicit a differential response in the organism, however on this second aspect there are no well-defined molecular bases but only statistical correlations and therefore an experiment format of " black box”. It is worth remembering that the quantitative analysis is based on the study of the 16s ribosomal DNA sequence, which is differentially repeated in tandem in bacterial species, and on the other hand, there are no data on differences in amplification efficiency in PCR of this sequence during diversity studies. This can lead to strong biases in the analysis of results. On the other hand, the constant variation of the microbiome must be understood from an evolutionary perspective, and therefore the homeostatic regulation of large organisms had to adapt to these changes, thereby avoiding major physiological alterations, and of course diseases. These arguments help explain the pattern that is normally found in all the articles that address the analysis of the microbiome considering the term dysbiosis, that is, that change is constant. This comment needs to be considered by the researchers, since the terms are assumed instead of questioned and this affects the research questions and the inference of results. However, there are multiple studies (some are cited in this work) that emphasize the relevance of the molecular bases of the colonization process in mammals associated with the development of the immune system; however, in my opinion, it is a situation with parameters that are not very comparable with adult individuals in terms of induction of differential genetic regulation.

On the other hand, in another part of the text it says “Research in the past two decades indicated disorders of altered GB interactions as well as psychiatric and neurological disorders such as depression, anxiety, AD, Parkinson's disease (PD), and autism spectrum disorder (ASD) [26].”, in this sense, it is assumed that the etiology of the disease is associated with the microbiome modification, however the molecular bases of these works from my point of view are debatable, since the behavioral changes Associated with these pathologies they also involve changes in the factor that most affects the microbiome composition , the diet, and therefore it is not sufficiently clear if it is the pathology that induces the perceived change in the microbiome of the patients. A situation in which the organism has not developed homeostatic regulation in the face of an environmental change that results in a modification of the microbiome, would inevitably lead to a sick population... which does not happen

The examples included in the work of secondary metabolites that are produced by bacteria with the potential to induce different physiological mechanisms are appropriate and interesting; however, in these works the activation of attenuation or repression mechanisms of a response induced by this type of metabolites that will be consistent with the thesis that I previously explained. The work says “The alteration of the gut microbiome in DS patients has also been reported [40-42]. Recent papers have indicated, that pregnant mothers, delivering babies with DS showed an altered microbiome compared to the control group”, I understand that in the control group there was homogeneity in the sample?,

As for the graphs, it strikes me, since despite the fact that a correlation is indicated in the graph, none can be seen, the dispersion of the points does not allow, in my opinion, to correlate the two variables. Some trend line can be seen, but no correlation. I understand that these analyzes include a lot of diversity in the data, however in addition to the lack of correlation, they did not consider differences in diet, which is the main input of bacteria to the microbiome and therefore directly affects the composition of the microbiome. I think that for his work to be accepted it is necessary to eliminate the categorical expressions about the correlation and assume that the analyzes in the microbiome are intrinsically variable and must be understood from an evolutionary perspective associated with Microbial Ecology more than with the clinic.

Author Response

Thank you very much for the reviewer’s work and the useful observations. My responses are inserted below the paragraphs.

From the scientific point of view, this work provides a statistical analysis where it is possible to establish a correlation between the consumption of antibiotics with a molecular mechanism associated with the disruption of the microbial cell wall synthesis process. However, there are several elements that must be considered

Our comparative analysis is based on three different statistical methods showing the same, significant correlation between the uses (consumption) of tetracycline (J01A), narrow spectrum, beta-lactamase resistant penicillin (J01CF) and the prevalence of people living with DS in 30 European countries. It can be considered a strong argument for this association.

The differences found between the prevalence between countries is an element to consider, since it would make it possible to identify a sample with the genetic, epigenetic and phenotypic characteristics that could contribute to identifying the causes and molecular bases that allow to delve into the disease etiology. Although this question is alluded to in the work “There are well-documented, classic events in the pathogenesis of the DS like maternal age, chromosome nondisjunction, and influence of paternal factors, all of which are results of early DS research. Unanswered questions have also been addressed by methods in the recent genomic medicine era, like robust genome-wide association studies which have been performed to identify the genomic susceptibility factors behind the mother’s predisposition. However, it is still clear that there are numerous factors outside of the human genome influencing DS susceptibility in pregnancy”. I understand that it should be properly referenced

I think, this issue is appropriately referenced.

One of the strong criticisms that must be made of this type of work refers to the term dysbiosis, which alludes to an abnormal qualitative and quantitative configuration of the microbiome. In this sense, it is worth reminding the authors that this term It is clinical and from the point of view of Microbial Ecology it does not make sense since the variation of species and the quantitative fluctuations in the microbiome are constant, therefore, there is not a situation of absence of dysbiosis, but rather there is a gradient of variation that is constant and more or less intense. This variation is supposed to have as a final result the production of certain secondary metabolites that elicit a differential response in the organism, however on this second aspect there are no well-defined molecular bases but only statistical correlations and therefore an experiment format of " black box”. It is worth remembering that the quantitative analysis is based on the study of the 16s ribosomal DNA sequence, which is differentially repeated in tandem in bacterial species, and on the other hand, there are no data on differences in amplification efficiency in PCR of this sequence during diversity studies. This can lead to strong biases in the analysis of results. On the other hand, the constant variation of the microbiome must be understood from an evolutionary perspective, and therefore the homeostatic regulation of large organisms had to adapt to these changes, thereby avoiding major physiological alterations, and of course diseases. These arguments help explain the pattern that is normally found in all the articles that address the analysis of the microbiome considering the term dysbiosis, that is, that change is constant. This comment needs to be considered by the researchers, since the terms are assumed instead of questioned and this affects the research questions and the inference of results. However, there are multiple studies (some are cited in this work) that emphasize the relevance of the molecular bases of the colonization process in mammals associated with the development of the immune system; however, in my opinion, it is a situation with parameters that are not very comparable with adult individuals in terms of induction of differential genetic regulation.

On the other hand, in another part of the text it says “Research in the past two decades indicated disorders of altered GB interactions as well as psychiatric and neurological disorders such as depression, anxiety, AD, Parkinson's disease (PD), and autism spectrum disorder (ASD) [26].”, in this sense, it is assumed that the etiology of the disease is associated with the microbiome modification, however the molecular bases of these works from my point of view are debatable, since the behavioral changes Associated with these pathologies they also involve changes in the factor that most affects the microbiome composition , the diet, and therefore it is not sufficiently clear if it is the pathology that induces the perceived change in the microbiome of the patients. A situation in which the organism has not developed homeostatic regulation in the face of an environmental change that results in a modification of the microbiome, would inevitably lead to a sick population... which does not happen

The role of dysbiosis in the pathogenesis of certain disease are debated by the honorable Reviewer. Dysbiosis, associated with the occurrence of different neurodegenerative, neurodevelopmental, metabolic, intestinal diseases and even malignancies are well documented in the relevant literature and it does not contradict the opinion of the Reviewer that the microbiome alters trough out the life as adapting to environmental circumstances including food intake (diet), but in an average European population, even within a country, disease-related dysbiosis might occur among the people eating similar diet. In this cases the microbiome alteration develops as a result of other external factors, like antibiotic consumption/pollution (or something else). The pathogenic role of dysbiosis is well documented in the publications when fecal microbial transplantation (FMT) was applied to ameliorate the symptoms of different diseases (Pakinson’s disease, autism, ulcerative colitis, etc.). Those observations contradict the remarks of the Reviewer.

(Vendrik KEW, Ooijevaar RE, de Jong PRC, Laman JD, van Oosten BW, van Hilten JJ, Ducarmon QR, Keller JJ, Kuijper EJ, Contarino MF. Fecal Microbiota Transplantation in Neurological Disorders. Front Cell Infect Microbiol. 2020 Mar 24;10:98. doi: 10.3389/fcimb.2020.00098. PMID: 32266160; PMCID: PMC7105733.)

The examples included in the work of secondary metabolites that are produced by bacteria with the potential to induce different physiological mechanisms are appropriate and interesting; however, in these works the activation of attenuation or repression mechanisms of a response induced by this type of metabolites that will be consistent with the thesis that I previously explained. The work says “The alteration of the gut microbiome in DS patients has also been reported [40-42]. Recent papers have indicated, that pregnant mothers, delivering babies with DS showed an altered microbiome compared to the control group”, I understand that in the control group there was homogeneity in the sample?,

The publication (42) clearly indicated the difference in microbiome between the mothers delivering babies with DS and the mothers delivering healthy babies. The DS group and the healthy control group of mothers were homogenous.

As for the graphs, it strikes me, since despite the fact that a correlation is indicated in the graph, none can be seen, the dispersion of the points does not allow, in my opinion, to correlate the two variables. Some trend line can be seen, but no correlation. I understand that these analyzes include a lot of diversity in the data, however in addition to the lack of correlation, they did not consider differences in diet, which is the main input of bacteria to the microbiome and therefore directly affects the composition of the microbiome. I think that for his work to be accepted it is necessary to eliminate the categorical expressions about the correlation and assume that the analyzes in the microbiome are intrinsically variable and must be understood from an evolutionary perspective associated with Microbial Ecology more than with the clinic.

The scatter diagram is generated by the computer after uploading the extensive databases what we have used for comparison, and the upward trend line indicates the (significant) association between the higher consumption tetracycline (J01A), the narrow-spectrum, beta-lactamase resistant penicillin (J01CF) and the prevalence of people living with DS. The significance is calculated/confirmed by three different statistical methods. This article focuses attention on the clinical aspects of this association and does not elaborate on microbial ecology. In this aspect, agreeing to the role of the diet effecting the composition of the microbiome, I wish to stress, that within a country’s population, probably eating similar food, there are other factors also influencing the components of the intestinal microbial taxa, not only the diet.

Reviewer 2 Report

The authors studied the correlation/association between overall antibiotic consumption in 30 European countries and the percentage of population living with Down Syndrome (DS) in those countris. While this correlation is interesting, this study is not enough to say that DS is associated with dysbiosis due to antibiotic use. As the authors have described in the introduction and discussion of the manuscript. DS occurrence likely involves multiple complex factors much of which is not yet understood. This study does not include any individual level data or any data on affects of antibiotic consumption on persons with DS or children born with DS. While the data analysis is robust, the conclusions drawn by the authors from the data are two broad and overreaching.  The title should be changed to reflect the study and the findings of the study, which is simply that there is a statistical correlation with greater consumption of tetracycline and higher percentage of DS in populations. It cannot be concluded or suggested from this study that there is an association between higher tetracycline consumption leading to DS occurrence. This would be an interesting line of investigation but much more work is required to suggest this.

Author Response

Thank you very much for the reviewer’s work and the useful observations. My response is inserted below the paragraph.

The authors studied the correlation/association between overall antibiotic consumption in 30 European countries and the percentage of population living with Down Syndrome (DS) in those countries. While this correlation is interesting, this study is not enough to say that DS is associated with dysbiosis due to antibiotic use. As the authors have described in the introduction and discussion of the manuscript. DS occurrence likely involves multiple complex factors much of which is not yet understood. This study does not include any individual level data or any data on effects of antibiotic consumption on persons with DS or children born with DS. While the data analysis is robust, the conclusions drawn by the authors from the data are two broad and overreaching.  The title should be changed to reflect the study and the findings of the study, which is simply that there is a statistical correlation with greater consumption of tetracycline and higher percentage of DS in populations. It cannot be concluded or suggested from this study that there is an association between higher tetracycline consumption leading to DS occurrence. This would be an interesting line of investigation but much more work is required to suggest this.

Our comparative analysis is based on three different statistical methods with the same results to support our observation between the association of the consumption of tetracycline (J01A), the narrow spectrum, beta-lactamase resistant penicillin (J01CF) and the prevalence of people living with DS in 30 European countries. At the same time we have clearly indicated at the end of the article: Limitations of our study: Our results could not be applied at the individual level, and we could not provide data on the molecular mechanism of how antibiotics act in the development of trisomy.

In our article we have only attempted to find explanation for our findings, which is clearly based on strong statistical evidences, and we have raised the possibility to a similar association between the consumption of certain antibiotics, the possible alteration of the gut flora and the occurrence of different neurodevelopmental disorders (and other diseases) extensively published in the literature.

The title of the article has been modified, as proposed by the honorable Reviewer.

Round 2

Reviewer 1 Report

In reference to your responses and understanding your line of research, I have to say that your responses do not articulate the responses to the arguments that I presented in my first comments. In many articles and even in the reference that was included in the document, the hypotheses are analyzed as "potential effects", because it is admitted that they are not well characterized, I would say that the molecular bases are non-existent. From my point of view, statistical analysis does not allow establishing a correlation, since a trend line is drawn with a very low coefficient, and a very high degree of dispersion. Finally, regarding your comment referring to the fact that it is a work focused on human health and not on Microbial Ecology, it is precisely a systematic error to treat a clearly bacterial ecosystem from the clinic without the considerations that affect microbial ecosystems. On the other hand, this and other studies address these approaches without considering the homeostasis that had to be developed by organisms in the face of constant change in microbiomes. This is a question that is never addressed because it has no clinical explanation. With these considerations it is not possible for me to accept this work.

Author Response

XXXCCCCV

Thank you for your observations, and I regret that my reasoning are not considered appropriate from your point of view. I wish to put forward some more arguments for your considerations:

  1. Our only statement in this article is that we found statistical associations between the consumption patterns of certain classes of antibiotics (tetracycline, penicillin) and the prevalence of people living with DS in 30 European countries.
  2. The article did not indicate the finding of the sole reason, the molecular basis, of the chromosome 21 nondisjunction, but we raise the possibility of this association.
  3. The importance of the adaptive microbial flora is clearly referred in the text as: Even though microbiome might vary among healthy individuals, microbial functions are well-conserved and associated with the generation of microbial metabolites which influence both microbes and hosts.
  4. Statistical tests were performed using IBM SPSS 26 software package. Pearson correlation was applied to estimate the correlation between antibiotic consumption and the preva-lence of DS. A significant correlation (positive/negative) was considered when p values were ≤ 0.05. A non-significant correlation was estimated when the p values fall between 0.051-0.09. The results are summarized in the paragraph of “Results” and at the end of Table 1. Doubting the statistical results, calculated automatically after uploading large databases, could not be a point of argument.
  5. The scatter diagram does not include the statistical results! To reduce the “dispersion of points”, after arranging the data in a rank order, results the same (upward) trend line.
  6. The opinion of the Honorable Reviewer that certain changes of the microbiome, apart from the physiological adaptation to maintain homeostasis, could not be implicated as an etiological factor in certain human diseases, contradicts the results of several observations, and experimental results which, in my point of view, could not be disregarded either.

Reviewer 2 Report

The authors addressed my concerns adequately.

Author Response

Thank you for accepting the answers.

Round 3

Reviewer 1 Report

In reference to your last comments, I understand the justification you make of the work, from my point of view the arguments that I included in my first comments have not been answered rigorously enough, however, it must be recognized that it is not something that can be circumscribe this work but is a common line in this type of study.

Only, and in reference to your last comment, where reference is made to studies that indicate that there is no homeostatic regulation that prevents certain diseases from emerging, I completely disagree, despite the fact that these studies provide indications, the establishment of postulates that remain in time in the scientific literature requires explaining from the evolutionary perspective any phenomenon in biological systems. There is no work that explains why the natural variation of the microbiome in people results in the emergence of large masses of diseased populations. Determining the etiology of a disease in a statistical analysis resulting from a confluence of data from a microbiome, whose complexity cannot currently be analyzed mathematically, lacks scientific rigor in my opinion.

In any case, and assuming that we are not going to reach an agreement, I am going to recommend to the editor that it be accepted with minor corrections and that they avoid sending me the manuscript again. Dysbiosis is not an acceptable term from a scientific perspective since it assumes a non-existent ecosystem reality.